



# Evaluating the contribution of the unexplored photochemistry of aldehydes on the tropospheric levels of molecular hydrogen (H₂).

Maria Paula Pérez-Peña[1], Jenny A. Fisher[2], Dylan B. Millet[3], Hisashi Yashiro[4], Ray L. Langenfelds[5], Paul B. Krummel[5], and Scott H. Kable[1]

[1]School of Chemistry, University of New South Wales, Sydney, NSW, Australia
[2]Centre for Atmospheric Chemistry, School of Earth, Atmospheric and Life Sciences, University of Wollongong, Wollongong, NSW, Australia
[3]Department of Soil, Water and Climate. University of Minnesota, Saint Paul, MN, USA
[4]Earth System Division, National Institute for Environmental Studies, Tsukuba, Japan
[5]Climate Science Centre, CSIRO Oceans & Atmosphere, Aspendale, Australia

**Correspondence:** Jenny A. Fisher (jennyf@uow.edu.au). Maria Paula Pérez-Peña (m.perez_pena@unsw.edu.au)

**Abstract.** Molecular hydrogen, H₂, is one of the most abundant trace gases in the atmosphere. The main known chemical source of H₂ in the atmosphere is the photolysis of formaldehyde and glyoxal. Recent laboratory measurements and ground-state photochemistry calculations have shown other aldehydes photo-dissociate to yield H₂ as well. This aldehyde photochemistry has not been previously accounted for in atmospheric H₂ models. Here, we used two atmospheric models to test the implications

5    of the previously unexplored aldehyde photochemistry on the H₂ tropospheric budget. We used the AtChem box model implementing the nearly chemically explicit Master Chemical Mechanism at three sites selected to represent variable atmospheric environments: London, Cape Verde and Borneo. We conducted five box model simulations per site using varying quantum yields for the photolysis of 16 aldehydes and compared the results against a baseline. The box model simulations showed that the photolysis of acetaldehyde, propanal, methylglyoxal, glycolaldehyde and methacrolein yield the highest chemical produc-

10    tion of H₂. We also used the GEOS-Chem 3-D atmospheric chemical transport model to test the impacts of the new photolytic H₂ source on the global scale. A new H₂ simulation capability was developed in GEOS-Chem and evaluated for 2015 and 2016. We then performed a sensitivity simulation in which the photolysis reactions of six aldehyde species were modified to include a 1% yield of H₂. We found an increase in the chemical production of H₂ over tropical regions where high abundance of isoprene results in the secondary generation of methylglyoxal, glycolaldehyde and methacrolein, ultimately yielding H₂. We

15    calculated a final increase of 0.4 Tg yr$^{-1}$ in the global chemical production budget, compared to a baseline production of ∼41 Tg yr$^{-1}$. Ultimately, both models showed that H₂ production from the newly discovered photolysis of aldehydes leads to only minor changes in the atmospheric mixing ratios of H₂, at least for the aldehydes tested here when assuming a 1% quantum yield across all wavelengths. Our results imply that the previously missing photochemical source is a less significant source of model uncertainty than other components of the H₂ budget, including emissions and soil uptake.





# 1   Introduction

The current global climate crisis has prompted governments to take actions towards decreasing greenhouse gas emissions. Countries like Australia (COAG, 2019), Germany (BMWi, 2020) and England (UK Secretary of State for Business, 2021) have announced plans to migrate from fossil fuels to use other energy carriers, including molecular hydrogen, $H_2$. These plans, some recent as 2020, have sparked a renewed interest in the so-called hydrogen economies. There are several advantages to the use of $H_2$ as fuel, most importantly, it can facilitate reaching carbon neutrality (van Renssen, 2020). Because of this potential to help tackle carbon neutral goals $H_2$ production from physical and chemical sources and the role of $H_2$ in tropospheric chemistry have been widely studied. However, new findings suggest that there is a previously unaccounted chemical source of $H_2$: direct production from the photolysis of a range of aldehydes in the atmosphere (Rowell et al., 2021). Here, we use atmospheric models to evaluate the impact of this unexplored source on the budget of tropospheric $H_2$.

Much of our understanding of the tropospheric $H_2$ budget comes from atmospheric models. Ehhalt and Rohrer (2009) reviewed extensively the available research on $H_2$ with particular focus on $H_2$ atmospheric modelling. The published work prior to Ehhalt and Rohrer (2009) focused mainly on understanding the atmospheric sinks and sources of $H_2$. The main known atmospheric sinks of hydrogen are the reaction with the hydroxyl radical, OH, and uptake by soil. The OH sink was estimated to account for approximately 24% of the total atmospheric sink, with the uptake by soil responsible for the remaining 76% of the loss Ehhalt and Rohrer (2009). The atmospheric lifetime of $H_2$ has been estimated to range between 1.4 years (Rhee et al., 2006; Xiao et al., 2007) to 2.3 years (Sanderson et al., 2003).

Sources of atmospheric $H_2$ are both primary, from combustion sources, and secondary, from the photolysis of volatile organic compounds, VOCs. Although modelling studies differ with respect to the magnitude of photochemical production of $H_2$ from VOCs (Novelli, 1999; Hauglustaine and Ehhalt, 2002; Sanderson et al., 2003; Rhee et al., 2006; Price et al., 2007; Xiao et al., 2007; Ehhalt and Rohrer, 2009; Yver et al., 2010; Yashiro et al., 2011; Derwent et al., 2020), they agree that this source accounts for at least 50% of the total, with the remaining percentage attributable to direct emissions.

Formaldehyde, HCHO, and glyoxal, $C_2H_2O_2$, are the two VOCs that produce $H_2$ once photolyzed. However, HCHO is considered to be the dominant photochemical source of $H_2$ in the atmosphere. The photochemistry of HCHO has been widely explored for many decades (Fried et al., 1997; Pope et al., 2005); pressure and temperature-dependent quantum yields are available, and rate coefficients with many atmospheric oxidants (OH, $HO_2$, $O_3$, Cl and others) have been measured (Burkholder et al., 2015).

In recent years, evidence has emerged that $H_2$ is also a primary photolysis product of other carbonyls. Harrison et al. (2019) found that $H_2$ is directly produced from the photolysis of acetaldehyde with a quantum yield of 1% at 1 atm and 298 K. Kharazmi (2018) found that longer carbonyls with $\beta-H$ atoms, like propanal and methylpropanal, had even higher, though still modest, quantum yields of 3% and 8% respectively. These discoveries gave rise to the hypothesis that photolysis of all aldehydes might also directly yield $H_2$. To test this hypothesis, Rowell et al. (2021) performed calculations on the photolysis pathways of many aldehydes, including the most atmospherically relevant ones, and provided theoretically estimated thresholds for the photo-dissociation channels that can form $H_2$. These calculations show that $H_2$ can be produced by several





mechanisms, depending on the chain length of the aldehyde. Aldehydes with at least two carbon atoms can release $H_2$ via a direct elimination pathway with a ketene co-fragment. Aldehydes with three or more carbons atoms in a chain can fragment concertedly into three fragments (known as triple fragmentation): $H_2$, CO and an alkene. These pathways are available in both saturated, aliphatic aldehydes (such as acetaldehyde and propanal) and unsaturated olefinic aldehdydes (such as acrolein and methacrolein). Molecules with different side chains, such as glycolaldehyde, also had $H_2$ elimination pathways.

In this work, we explore the implication of this direct generation of $H_2$ from aldehydes on atmospheric chemistry by implementing the nearly explicit Master Chemical Mechanism (hereafter MCM) in a box model (AtChem v 1.2) and updating the 3-D global atmospheric chemical transport model GEOS-Chem v12.5.0 (DOI:10.5281/zenodo.3403111) to include an $H_2$ simulation. Neither model in its default configuration currently includes the direct photochemical production of $H_2$ from any aldehydes except formaldehyde and glyoxal. The box modelling with MCM allowed us to test whether there could be an im-

pact from the unexplored photochemistry in a detailed chemistry scheme. The modelling with GEOS-Chem enabled us to test the repercussions on the global tropospheric $H_2$ budget while using a simplified chemistry scheme (*i.e.*, a reduced number of aldehydes). To enable the global modelling experiments, a standard baseline model of $H_2$ within GEOS-Chem was also developed here. An $H_2$ simulation capability was present in an early version of GEOS-Chem v5.05 (Price et al., 2007) but was not maintained in more recent model versions. Recent versions of the model have not included $H_2$ as a chemically active species,

assuming a fixed background value of 500 ppbv throughout the troposphere. Our addition of an $H_2$ simulation capability to GEOS-Chem v12.5.0 builds on the original implementation of Price et al. (2007) with improvements to both source and sink terms and will enable future GEOS-Chem studies of $H_2$.

This paper is divided into three parts. Section 2 summarizes the use of the box model with MCM and the experimental design to test the photochemistry of the aldehydes that can produce $H_2$. This section describes the model configuration, application,

and the results of adding new photochemical pathways to the chemical mechanism. Section 3 describes the GEOS-Chem modelling. It includes a description of the construction of a baseline $H_2$ simulation and its evaluation against a global ensemble of atmospheric $H_2$ observations. This section also describes the global implications of the production of $H_2$ from the photolysis of aldehydes. Finally, Section 4 provides a summary and the conclusions derived from this work.

## 2 Box modelling implementing the Master Chemical Mechanism

### 2.1 Configuration of the box model simulations of $H_2$

The current version of the mechanism, MCM v3.3.1, accounts for the degradation of 142 volatile organic species and involves around 17,000 elementary reactions (Rickard and Young, 2018). For this work, MCM v3.3.1 was implemented in the open source box model AtChem V1.2 (Sommariva et al., 2020). The direct production of $H_2$ from the photolysis of aldehydes was tested with box model simulations for three sites. The sites chosen for the box modelling represent urban, pristine oceanic,

and pristine forested environments. For each site, the model was configured with different subsets of the MCM that were downloaded for selected species based on the measurements available to constrain the box model. Likewise, the length of each simulation was set based on the dates for which measurements were available to constrain the box model.





Table 1 contains a summary of the box model simulations. The first box model simulation was run for London, with 71 chemical species constrained by the measurements from the ClearfLo (Clean Air for London) campaign of 2012 (Bohnensten-

gel et al., 2015). The measurements to constrain the box model for London were previously used by Shaw et al. (2018) to test the formation of formic acid from the photo-tautomerization of acetaldehyde. The box model for London considered 11,667 reactions for 3,880 species. The second model simulation was configured for Cape Verde. The measurements used to constrain the model were those from the Cape Verde Atmospheric Observatory, located in the tropical Atlantic marine boundary layer. A total of 12 species were constrained from measurements taken in January 2015 (Kozlova et al., 2019; Read, 2021a, b). This

box model set-up included 2,753 reactions for 894 species. The the third box model simulation was conducted for the southeast Asian tropical rainforest in Borneo. The measurements of 14 species from the OP3-III campaign at Bukit Atur Sabah, Malaysia (Hewitt et al., 2010) were used as constraints. For the third box model simulation, the subset of the MCM contained 4,196 reactions for 1,356 species. All three models included a total of 18 aldehydes (see Table S1) with their corresponding reactions. Of these, only two (formaldehyde and glyoxal) already included $H_2$ as a photolysis product in the standard MCM

v3.3.1 mechanism.

An initial baseline using the standard MCM v3.3.1 was simulated to use for later comparison against four sets of sensitivity tests. All simulations were initialized with an $H_2$ abundance of $1.30 \times 10^{13}$ molecules cm$^{-3}$ ($\sim$530 ppbv). The value was chosen based on the average $H_2$ mixing ratios from four measuring sites located across the world (Krummel et al., 2021b, a, d, h). The four sensitivity tests were designed to explore the relative contribution that the photolysis of aldehydes could have on the

chemical production and resultant mixing ratios of $H_2$. To that end, new photochemical channels were added for 16 aldehydes available in the extracted MCM subsets (see Table S1), with $H_2$ specified as a primary product. Previous experimental findings have demonstrated $H_2$ quantum yields that varied from 1% for acetaldehyde (Harrison et al., 2019) to 8% for methylpropanal (Kharazmi, 2018). Consequently, the four box model tests used uniform 1%, 2%, 5% and 10% quantum yields across all 16 available aldehydes and across all wavelengths, with the goal of bracketing the plausible range of behaviour. We did not

include physical processes in the box modelling, as the goal of these simulations was to explore the contribution to $H_2$ from aldehyde photolysis relative to the other chemical sources and sinks. This means that the box models do not consider either direct emissions or the soil uptake sink of $H_2$. Furthermore, the soil uptake is not relevant at this stage because of the timescale in which we conducted the simulations.

The individual photolysis rates for the new aldehyde photolysis reactions were calculated and incorporated into each box

model simulation as constrained values at each site. The calculation of the photolysis rates ($J$) for each aldehyde followed the Equation 1:

$$J = \int_{\lambda 1}^{\lambda 2} F(\lambda)\phi(\lambda)\sigma(\lambda)\,d\lambda \qquad (1)$$

where $\lambda 1$ is 290 nm, $\lambda 2$ is 345 nm, $F$ is the actinic flux as a function of the zenith angle, $\sigma$ is the cross section of each aldehyde and $\phi$ is the quantum yield, varied as explained previously (1%, 2%, 5% and 10%). Table S1 lists the aldehydes available in





**Table 1.** Summary of the box model configurations.

| Site | Start date | End date | No. of reactions | No. of constrained species | Measurement constraints |
|------|-----------|----------|------------------|----------------------------|--------------------------|
| London | 22-Jul-2012 | 3-Aug-2012 | 11,667 | 71 [a] | ClearfLo |
| Cape Verde | 01-Jan-2015 | 12-Jan-2015 | 2,753 | 12 [b] | Cape Verde Atmospheric Observatory (Kozlova et al., 2019; Read, 2021a, b) |
| Borneo | 11-Jul-2008 | 17-Jul-2008 | 4,196 | 14 [c] | OP3-III (Hewitt et al., 2010; NCRE et al., 2010, 2009b, a) |

[a] 1−Butanal, 1−butene, 1−pentene, 1−propanol, 1,2−dimethylethylene, 1,3−butadiene, 2−butene, ethyne, 2−ethyltoluene, 2−hexanone, 2−methylbutane, 2−methylpentane, 2−methylpropanal, 2−pentanone, 3−ethyltoluene, 4−ethyltoluene, 4−methyl−2−pentanone, Acetaldehyde, Acetone, $\alpha$−pinene, Benzaldehyde, Benzene, Butanal, Butane, CO, Cyclohexanone, Decane, Dodecane, Ethane, Ethanol, Ethene, Ethyl acetate, Ethylbenzene, Formaldehyde, Hemimellitene, Heptane, Hexane, Isobutane, Isobutene, Isoprene, Isopropylbenzene, Limonene, M−xylene, Mesitylene, Methacrolein, Methane, Methanol, Methyl ethyl ketone, Methyl vynil ketone, Methylene chloride, Nitric acid, NO, $NO_2$, Nonane, O−xylene, Octane, Ozone, P−xylene, Pentanal, Pentane, Peroxyacetyl nitrate, Propane, Propene, Propylbenzene, Pseudocumene, Styrene, Toluene, trans−2−pentene, Trichloroethylene, Undecane, $H_2O$

[b] 2−methylbutane, Butane, Ethane, Ethene, Isobutane, Methane, Ozone, Pentane, Propane, Propene, Toluene, Benzene

[c] 1−butene, Acetaldehyde, Acetone, $\alpha$−pinene, Ethene, Ethyne, Formaldehyde, Limonene, Methacrolein, Methanol, $NO_2$, Isoprene, Ozone, Propene

MCM that were used in the model sensitivity experiments, along with the corresponding photolysis products and reaction rates calculated using Equation 1. For the reactions where $H_2$ is produced partnered to a ketene molecule, glycolaldehyde was used as a surrogate species for ketene as neither MCM nor GEOS-Chem include ketene in their mechanisms.

### 2.2   Box modelling results: contributions of aldehydes to the photochemical production of $H_2$

Because of the short modelled times (limited by available measurement constraints), none of the baseline simulations repre-
sented steady state conditions. As a result, we focus on interpretation of changes to chemical production rather than changes to mixing ratios. However, we first briefly describe mixing ratios in the baseline simulations to provide context for our results. Select modelled species from the baseline simulations for London, Cape Verde and Borneo are shown in Figures S1-S3 respectively. The baseline mixing ratio of $H_2$ increased continuously from its initial value of 530 ppbv to 639.97 ppbv at London and 533.96 ppbv at Borneo. At Cape Verde, the mixing ratio of $H_2$ decreased during the simulated period to 517.23 ppbv. These
changes are equivalent to an increase in $H_2$ of ∼17% over 12 days for London, an increase of ∼1% over 6 days for Borneo, and a decrease of ∼2% over 12 days for Cape Verde. These final $H_2$ mixing ratios are not representative of the actual values





over the selected locations particularly because of the lack other relevant physical processes such as emissions, transport and uptake by soil.

For the London and Borneo simulations, the $H_2$ increase over time in the baseline run was effectively caused by the photolysis of formaldehyde and glyoxal (the only $H_2$ sources in this simulation). The decrease in modelled $H_2$ at Cape Verde was a result of imbalances in the chemical sources and sinks in this regime. For the Cape Verde simulation, neither formaldehyde nor glyoxal had available measurement constraints (see Table 1); however, both were produced chemically, with formaldehyde production from the degradation of methane, ethene, propene, toluene and benzene and glyoxal production from ethene and toluene precursors (Stavrakou et al., 2009). These five precursor species were all constrained in the Cape Verde simulations (see Table 1, footnote b). Figure S2 shows the time series of selected species modelled for Cape Verde, including formaldehyde, which had an average modelled value of ∼800 pptv (∼2×10$^{10}$ molecules cm$^{-3}$). For comparison, Whalley et al. (2010) reported an average noon value of 328 pptv for their MCM simulations at Cape Verde during May-June 2007 (note that they do not report values for glyoxal). Considering that our modelled formaldehyde mixing ratios were higher than those reported previously by Whalley et al. (2010), and because of the formaldehyde and glyoxal lifetimes of a few hours, we conclude that our simulations included sufficient precursor concentrations and therefore that the decrease in $H_2$ in the baseline Cape Verde simulation implies that the available $H_2$ was consumed by OH more rapidly than it could be produced by formaldehyde and glyoxal, yielding an effective loss over the 12 days modelled.

$H_2$ chemical production provides a more realistic and useful outcome from the box model simulations. At all three modelled sites, the relative rate of $H_2$ chemical production in the sensitivity simulations increases relative to the baseline simulation and scales linearly with the quantum yield over the 1-10% range studied here (see Figures S4 and S5). This linearity makes it reasonable to interpolate the predicted MCM production rates simulated here for as-yet unmeasured aldehydes to whatever experimental quantum yield is ultimately determined.

We use the sensitivity simulations to evaluate the relative importance of each aldehyde. Figure 1 displays the relative daytime contribution of each newly-considered aldehyde to the total aldehyde-derived photolysis $H_2$ production rates modelled using a 1% quantum yield (not including the contributions from formaldehyde and glyoxal, which were not modified in this work). To aid the eye, aliphatic aldehydes are grouped in orange tones, unsaturated aldehydes (and methylglyoxal) in green tones and oxygenated aldehydes in blue tones. For the London site, aliphatic aldehydes (orange) dominated, with acetaldehyde by far the largest contributor (74%). The remaining contributions were distributed between propanal (12%), methylglyoxal (5%), glycolaldehyde (5%), and to a lesser extent acrolein and methacrolein. The contribution of biogenic-related species in London like methylglyoxal, glycolaldehyde and methacrolein can likely be attributed to the presence of isoprene, which reached a maximum value of ∼400 pptv (∼1.0×10$^{10}$ molecules cm$^{-3}$) as shown in Figure S1.

While six species contributed to the modelled production of $H_2$ from aldehydes at London, at the Cape Verde site just three aldehydes were dominant. Aliphatic aldehydes, mostly acetaldehyde (72%) and propanal (18%), again dominated $H_2$ production, with an additional contribution from methylglyoxal (10%). There was virtually no contribution from any other aldehyde.





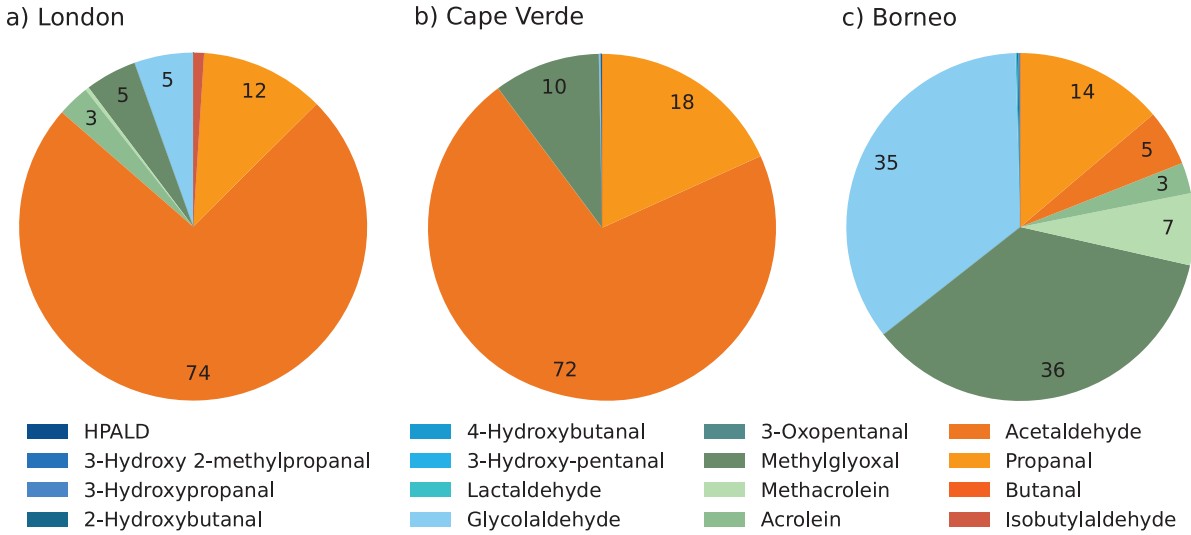

**Figure 1.** Average daytime percentage contribution of individual aldehydes to the total production of $H_2$ from aldehydes (excluding formaldehyde and gloyxal) assuming a 1% quantum yield, estimated for the a) London, b) Cape Verde and c) Borneo sites using the box model.

For Borneo, the modelled distribution of the aldehyde contributions to $H_2$ production was completely different than modelled at the other two sites. Aliphatic aldehydes provided only a minor contribution. There was a particularly notable difference in the influence of acetaldehyde, which represented more than half the new $H_2$ production at London and Cape Verde but only 5% of the production at Borneo. Meanwhile, the contributions of methylglyoxal (46%), glycolaldehyde (35%) and other unsaturated aldehydes were markedly larger at Borneo than at the other two sites. The results for Borneo clearly show the influence of biogenic isoprene in the rainforest atmosphere, as methylglyoxal, glycolaldehyde and methacrolein are all products of isoprene oxidation (Wennberg et al., 2018). The isoprene in Borneo reached values of up to 2350 pptv ($5.7 \times 10^{10}$ molecules cm$^{-3}$) (Hewitt et al., 2010) (see Figure S3). As a comparison, the isoprene mixing ratios in London were much lower at about ~17% of the Borneo values. No isoprene was simulated for Cape Verde, but previously reported typical noon values at Cape Verde of ~10 pptv (Whalley et al., 2010) imply its contribution to $H_2$ production from aldehydes there would be negligible. The larger effect of the new photochemical $H_2$ sources at Borneo relative to the other sites due to the abundance of biogenic VOCs implies the newly discovered photochemical production pathways will have most influence in biogenic source regions, and this will be explored in the next section using the global model.

The box modelling with the MCM v3.3.1 allowed us to test $H_2$ production from the photolysis of a wide range of aldehydes in a complex and explicit chemical mechanism. While $H_2$ did not reach steady state in any of the box models (due to the short simulation period), these box model simulations identified the aldehydes that are expected to contribute the most to photolytic production of $H_2$ under distinct environmental conditions. In urban environments, modelled as the London site, linear aliphatic aldehydes (especially acetaldehyde and propanal) are the most relevant. For regions with substantial vegetation (e.g., tropical forested areas such as Borneo), aldehydes that are produced from the oxidation of isoprene, such as methylglyoxal





and glycolaldehyde, are the most important. At all three sites, aside from glycolaldehyde, none of the oxygenated aldehydes modelled here (blue tones in Figure 1) featured with any significance to the formation of $H_2$. However, the short simulation times (driven by lack of appropriate observational constraints) and the absence of physical sources and sinks limit the usefulness of the box model results for further quantifying the effects of the relevant identified aldehydes on tropospheric photochemical formation of $H_2$. We therefore turned to a global chemical transport model (GEOS-Chem), in which we were able to include

not only the new photochemistry for the most relevant species as identified by the box modelling but also physical processes (emissions and soil uptake). With the global model, we were also able expand the evaluation to diverse environments across the globe and to run simulations for periods long enough to allow $H_2$ to reach steady state, providing more robust results. The global modelling of $H_2$ is described in the following section.

## 3 Global atmospheric modelling of $H_2$ using GEOS-Chem

### 3.1 GEOS-Chem model configuration: development of the baseline simulation of $H_2$.

GEOS-Chem is a widely used 3D chemical transport model originally described by Bey et al. (2001). The current work used version 12.5.0 modified to include $H_2$ as part of the standard chemistry simulation. To test the production of $H_2$ from aldehydes with the atmospheric model, we first constructed a baseline simulation that included all other known $H_2$ sources and sinks. All GEOS-Chem simulations were performed on a $4° \times 5°$ horizontal resolution with 72 vertical layers. The

simulations included stratospheric chemistry using the UCX mechanism (Eastham et al., 2014) and were driven by Goddard Earth Observing System Forward Processing (GEOS-FP) meteorology from the Global Modeling and Assimilation Office (GMAO). The baseline simulation was run from June 2014 to December 2016, with the first six months used as spin-up time. The default version of GEOS-Chem v12.5.0 does not include $H_2$ as an active species, and so the $H_2$ mixing ratio has a fixed concentration of 500 ppbv across the troposphere. However, observations compiled by CSIRO at four sites, two located in the

Northern Hemisphere (Krummel et al., 2021b, a) and two in the Southern Hemisphere (Krummel et al., 2021d, h) show that on average the global mixing ratio of $H_2$ is ∼530 ppbv. Based on these observations, the initial mixing ratio of $H_2$ was modified in GEOS-Chem to match the average observed value of 530 ppbv.

For our baseline configuration, we added known $H_2$ physical sources and sinks into GEOS-Chem. We scaled $H_2$ emissions to inventory estimates of carbon monoxide (CO) emissions as done previously in other studies (Ehhalt and Rohrer, 2009) as there

are no dedicated emission inventories available for $H_2$. The scaling was performed using the Harmonized Emissions Component (HEMCO) in GEOS-Chem (Lin et al., 2021). Two different emission ratios were implemented, one for anthropogenic combustion sources (0.042 g $H_2$/g CO) and the other for biomass burning sources (0.021 g $H_2$/g CO) based on the fractions used by Price et al. (2007). For anthropogenic $H_2$ emissions, the Community Emissions Data System (CEDS) global inventory (Hoesly et al., 2018) was used. The CEDS emissions were overwritten by more detailed regional emission inventories where

applicable: APEI for Canada, DICE-Africa for Africa (Marais and Wiedinmyer, 2016), EPA/NEI11 for North America and MIX for East Asia (Li et al., 2017). For the biomass burning $H_2$ emissions, the Global Fire Emissions Database v4 (GFED4) (Randerson et al., 2018) was used. Oceanic emissions were from Price et al. (2007), who distributed reference $H_2$ emissions


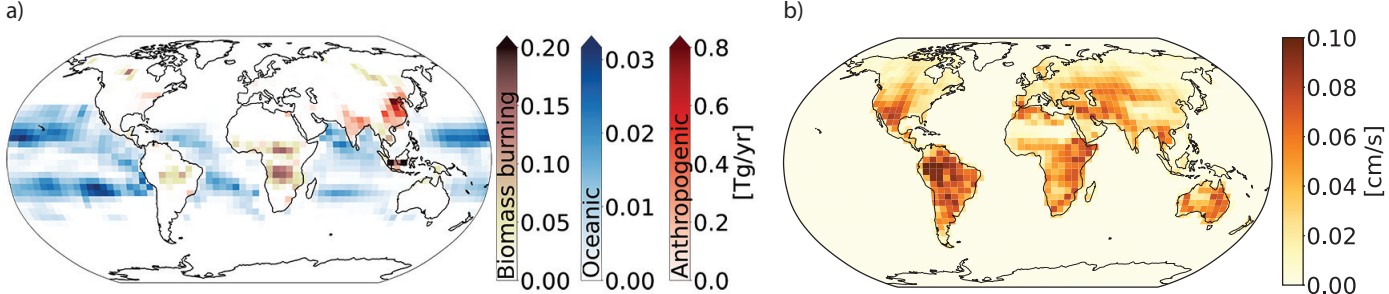

**Figure 2.** a) Average annual global emissions of $H_2$ in Tg yr$^{-1}$ from 2015 and 2016 from biomass burning (brown), oceanic (blue), and anthropogenic (red) sources. Anthropogenic emissions include fossil fuel and biofuel combustion. Note the difference in scales for the different source types. b) Average annual dry deposition velocity of $H_2$ in cm s$^{-1}$ supplied into GEOS-Chem and determined from the original calculations from Yashiro et al. (2011).

of 6 Tg yr$^{-1}$ globally using the spatial distribution of biological nitrogen fixation in the ocean determined by Deutsch et al. (2007). The simulation also uses the Model of Emissions of Gases and Aerosols from Nature, MEGANv2.1 (Guenther et al., 2012), for biogenic emissions of volatile organic compounds, several of which are the aldehydes that are included in this work as sources of $H_2$.

Figure 2a shows the average annual global primary sources of $H_2$ for 2015 and 2016 from biomass burning, oceanic and anthropogenic emissions (all in Tg yr$^{-1}$). The red pixels in Figure 2a show that China contributes the most to the emissions of $H_2$ due to the large anthropogenic sources there. Biomass burning emissions from the African savannas, Indonesia, the Amazon, and some parts of North America and Russia are important sources of molecular $H_2$ but to a lesser extent than the anthropogenic emissions. Oceanic $H_2$ sources show maximum emissions occurring over the Pacific with no emission at high latitudes (poleward of 40°S and 40°N), as described in detail by Price et al. (2007).

We also added the atmospheric $H_2$ sink from soil uptake. The soil uptake of $H_2$ involves both biological (enzymatic and microbial activity) and physical (molecular diffusion) processes, which jointly determine the magnitude of the sink (Yver et al., 2010). The correlation of the enzymatic and microbial activity with soil temperature and moisture drives the seasonality of atmospheric $H_2$. Soil temperatures between 20°C and 30°C are optimal to capture $H_2$, with no capture below −20°C or above 40°C. Likewise, arid and frozen soils have been shown to have low values of $H_2$ uptake (Yashiro et al., 2011). The rate of soil uptake of $H_2$ has been measured as a dry deposition velocity and thus is typically parameterised in models as a dry deposition process (Ehhalt and Rohrer, 2009; Yver et al., 2010; Yashiro et al., 2011). Reported values of dry deposition velocities of $H_2$ onto soils range from 0.01 to 0.15 cm s$^{-1}$ based on measurements in savanna, tundra and desert ecosystems and in agricultural lands (Ehhalt and Rohrer, 2009).

Although GEOS-Chem is capable of calculating the dry deposition velocities using a resistance-in-series scheme that relies on known parameters (including Henry's Law constants, reactivity factors, etc.), we instead used dry deposition velocities derived from a comprehensive soil $H_2$ model developed by Yashiro et al. (2011). Yashiro et al. (2011) used a two-layered





diffusion/uptake model that considers biologically inactive and active layers as well as the porosity and temperature of the soil. To our knowledge, Yashiro et al. (2011) provide the most comprehensive and highly resolved values of the dry deposition velocity of $H_2$, derived from a total of 8 modelled years (1997-2005) with daily resolution on a $1.25° \times 1.25°$ horizontal grid with global coverage. We used the dry deposition velocities from Yashiro et al. (2011) to create a climatology for an "average" year with daily temporal resolution, which was then used in our 2014-2016 simulations. The $1.25° \times 1.25°$ resolution dry

deposition velocities of $H_2$ were mapped to our modelled grid ($4° \times 5°$) using HEMCO.

Figure 2b shows the annual average $H_2$ dry deposition velocities gridded as used in GEOS-Chem. The dry deposition velocities ranged from 0.1 to 0.10 cm s$^{-1}$. Regions like the Sahara desert and far eastern Russia have the lowest dry deposition velocities because the uptake efficiencies are diminished by the extremely high or low soil temperatures and moisture content (Yashiro et al., 2011), which inhibit $H_2$ capture as described previously. Equatorial regions have the highest dry deposition

velocities, and in these regions they remain almost constant throughout the year.

Deposition onto water bodies is not considered in our simulations. Although Punshon et al. (2007) provide first-order net loss-rate constants of $H_2$ in seawater sampled in Canada, these values have not been extensively tested, and it is unclear whether they are broadly representative of other waters. Further, the loss rates are not well known under different conditions (e.g., salty tropical waters versus freshwater lakes). Ehhalt and Rohrer (2009) suggested that the deposition of $H_2$ onto water is at most a minor player in the tropospheric budget of $H_2$.

a minor player in the tropospheric budget of $H_2$. Considering this finding and the lack of extensive research on loss of $H_2$ to water bodies, we did not include this sink and expect this would have a negligible impact on the findings reported here.

Our baseline simulation also includes $H_2$ chemical production and loss. The standard chemical mechanism in GEOS-Chem already includes the major known chemical sources of $H_2$: photolysis of formaldehyde and glyoxal, reaction of excited oxygen atoms with methane, and reaction of the H atom with the hydroperoxyl radical. Similarly, the standard mechanisms also

includes the only known significant chemical $H_2$ sinks: reaction of $H_2$ with the hydroxyl radical and with chlorine atoms. While these sources and sinks were already present in GEOS-Chem v12.5.0, they did not influence simulated $H_2$ as it was set as a "fixed" species, with a constant value of 500 ppbv. Here we change $H_2$ to an active species so that the $H_2$ concentrations change in response to the chemical sources and sinks outlined above.

### 3.2    GEOS-Chem modelling results: evaluation of the baseline simulation

Before testing the impact of the new $H_2$ source from aldehyde photolysis, we first evaluated the performance of the baseline simulation. Table 2 summarizes the burden, lifetime and tropospheric budget of $H_2$ calculated for our baseline simulation, along with values from previous research. Using our new GEOS-Chem baseline configuration, we calculated the global burden of $H_2$ to be 159 Tg for 2015 and 157 Tg for 2016. These estimated values are within the range of previous reports of the global burden of $H_2$, which ranged from 136 Tg (Hauglustaine and Ehhalt, 2002) to 172 Tg (Sanderson et al., 2003). Our estimate for

the $H_2$ lifetime is 2 years, in agreement with previous reports that range from 1 year (Rhee et al., 2006; Xiao et al., 2007) to 2.3 years (Sanderson et al., 2003).

Compared against other studies, anthropogenic emissions are highest 23.7 Tg yr$^{-1}$ for 2015 and 23.8 yr$^{-1}$ for 2016. Our $H_2$ emissions inventory is based on more recent anthropogenic emissions than the previous studies (2015-2016 in our work versus





**Table 2.** Global tropospheric sources (Tg yr$^{-1}$), sinks (Tg yr$^{-1}$), burdens (Tg) and lifetimes (yr) of H$_2$

| | Novelli (1999) | Hauglustaine and Ehhalt (2002) | Sanderson et al. (2003) | Rhee et al. (2006) | Price et al. (2007) | Xiao et al. (2007) |
|---|---|---|---|---|---|---|
| Total emissions [a] | 37 | 39 | 48 | 43 | 39.8 | 28 |
| *Anthropogenic* [b] | 15±10 | 16 | 20 | 15±6 | 23.7 | 15±10 |
| *Biomass Burning* | 16±5 | 13 | 20 | 16±3 | 10.1 | 13±3 |
| *Biogenic N$_2$ fixation* [c] | 6±3 | 10 | 8 | 12±10 | 6 | |
| Chemical production | 40 | 31 | 30.2 | 64±12 | 34 | 77±10 |
| Total source | 77±16 | 70 | 78.2 | 107±15 | 73 | 105±10 |
| Soil uptake | 56±41 | 55 | 58.3 | 88±11 | 55±8.3 | 85±5 |
| Chemical loss | 19±5 | 15 | 17.1 | 19±3 | 18 | 18±3 |
| Total sink | 75±41 | 70 | 75.4 | 107±11 | 73 | 107±11 |
| Burden | 155±10 | 136 | 172 | 150 | 141 | 149±23 |
| Tropospheric lifetime | 2.1 | 1.9 | 2.3 | 1.4 | 2 | 1.4 |

[a] Includes anthropogenic (fossil fuel), biomass burning and biogenic N$_2$ fixation emissions.

[b] Includes fossil fuel and biofuel related emissions.

[c] Includes Land and Ocean N$_2$ fixation emissions.





**Table 2.** (continued) Global tropospheric sources (Tg yr$^{-1}$), sinks (Tg yr$^{-1}$), burdens (Tg) and lifetimes (yr) of $H_2$

| | Ehhalt and Rohrer (2009) | Yver et al. (2010) | Yashiro et al. (2011) | Derwent et al. (2020) | This work: Baseline[d] | This work: Scenario[d] |
|---|---|---|---|---|---|---|
| Total emissions[a] | 35 | 35.7±4.3 | 30-37 | 50 | 38.9 // 36.3 | 38.9 // 36.3 |
| *Anthropogenic*[b] | 11±4 | 18.5 | 15.1-15.4 | 20 | 23.7 // 23.8 | 23.7 // 23.8 |
| *BiomassBurning* | 15±6 | 7.8 | 8-15 | 20 | 9.2 // 6.5 | 9.2 // 6.5 |
| *BiogenicN$_2$fixation*[c] | 9±5 | 9.4 | 9 | 10 | 6 | 6 |
| Chemical production | 41±11 | 46.5±0.2 | 38-39 | 49 | 40.6 // 41.4 | 41.0 // 41.8 |
| Total source | 76±14 | 82.2±4.5 | 73-80 | 99 | 79.5 // 77.6 | 79.9 // 78.1 |
| Soil uptake | 60±30 | 58.8±9.0 | 57-60±12 | 79 | 59.7 // 59.7 | 59.9 // 59.9 |
| Chemical sink | 19±5 | 18.2±0.4 | 17-18 | 26 | 19.8 // 20.2 | 19.8 // 20.3 |
| Total sink | 79$^{+30}_{-20}$ | 77±9.4 | 75-78 | 105 | 79.5 // 79.9 | 79.6 // 80.2 |
| Burden | 155±10 | 166 | 148-153 | 150 | 159.0 // 157.5 | 159.5 // 158.3 |
| Tropospheric lifetime | 2 | 2.2 | 1.9-2.0 | 1.5 | 2.0 | 2.0 |

[a]Includes anthropogenic (fossil fuel), biomass burning and biogenic $N_2$ fixation emissions.

[b]Includes fossil fuel and biofuel related emissions.

[c]Includes Land and Ocean $N_2$ fixation emissions.

[d]Budgets are reported for 2015 // 2016.





2010 or earlier in most previous work). Biomass burning emissions in our simulations were amongst the lowest estimates,

with 9.2 Tg yr$^{-1}$ for 2015 and 6.5 Tg yr$^{-1}$ for 2016, compared to other estimates that ranged from 8 Tg yr$^{-1}$ (Yashiro et al., 2011) to 20 Tg yr$^{-1}$ (Derwent et al., 2020). As we used the same H$_2$ to CO emission ratio as in previous studies to derive the H$_2$ biomass burning emissions, we expect the discrepancy comes either from interannual variability between the different modelled years or from differences in the underlying CO emissions inventories. For the latter, most studies did not specify the inventory used for biomass burning; however previous work has shown that there can be large differences between inventories,

including GFEDv4s as used here (Desservettaz et al., 2021; Liu et al., 2020). Ocean emission estimates are within the reported values at 6 Tg yr$^{-1}$.

Chemical production of H$_2$ in our baseline simulation was 40.6 Tg yr$^{-1}$ for 2015 and 41.4 Tg yr$^{-1}$ for 2016, within the range from most recent studies of 30 Tg yr$^{-1}$ (Sanderson et al., 2003) to 49 Tg yr$^{-1}$ (Derwent et al., 2020). Earlier estimates of 64±12 yr$^{-1}$ from Rhee et al. (2006) and 77±10 Tg yr$^{-1}$ from Xiao et al. (2007) are higher than all other estimates, a difference

that Ehhalt and Rohrer (2009) have attributed to their use of top-down inverse methodology, as opposed to the bottom-up approach used in other studies (including our baseline). The generally good agreement in the H$_2$ chemical source between our baseline and the other studies indicates that photolysis of formaldehyde and glyoxal yields H$_2$ production consistent with prior estimates, providing an appropriate baseline to compare to the so far unexplored photochemical production of H$_2$ from other aldehydes.

As in previous work, the soil uptake sink was almost three times higher than the chemical sink in our baseline simulation. We simulated a soil uptake sink of ∼60 Tg yr$^{-1}$ which fell within the range reported by Yashiro et al. (2011), the source of the H$_2$ dry deposition velocities used here. Most other studies also show a similar soil uptake sink, with a few exceptions as described in detail by Yashiro et al. (2011). The chemical sink in our baseline simulation was ∼20 Tg yr$^{-1}$, again consistent with all other studies. The strength of the soil uptake sink varied throughout the year, while the chemical sink was largely stable

(Figure 3 c and d). In general, the tropospheric burden, budget and lifetime of H$_2$ determined using our baseline compared well with previous references.

We evaluated the modelled mixing ratios of H$_2$ from the baseline simulation using CSIRO measurements reported by Krummel et al. (2021b, c, d, e, f, g, h, a). The CSIRO datasets correspond to monthly flask air sample measurements of H$_2$ (along with CO$_2$, CH$_4$, CO, N$_2$O, and $^{13}$C and $^{18}$O isotopes of CO$_2$) at 8 ground based sites with data from 1992 to 2019 and all

include measurements for 2015 and 2016, plus one set of H$_2$ aircraft measurements with data from 1991 to 2000. Six of the ground sites in this dataset are located in the Southern Hemisphere, with the remaining two sites in the Northern Hemisphere. CSIRO data are reported in the MPI-2009 scale (Jordan and Steinberg, 2011).

Previous work has compared modelled H$_2$ against measurements by the National Oceanic and Atmospheric Adminis- tration (NOAA)/Earth System Research Laboratory (ESRL) (Price et al., 2007; Ehhalt and Rohrer, 2009; Yashiro et al., 2011).

The NOAA dataset provides around 125 measuring sites with H$_2$ measurements across the world starting from 1989 (Dlugokencky et al., 2017). However, the NOAA data are subject to calibration issues that remain unresolved (Masarie et al., 2001). CSIRO data, supported by a more robust calibration scheme, are used for this study despite having lower spatial coverage of measurement sites.





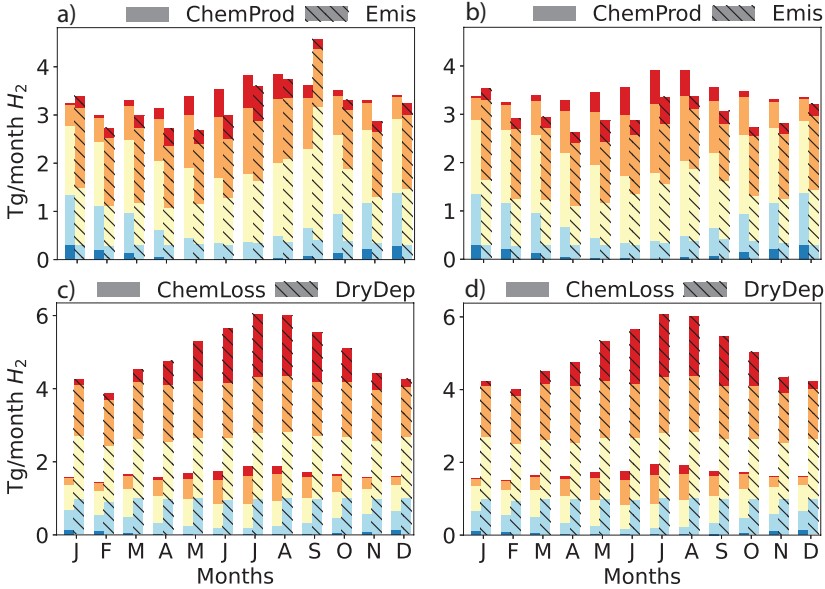

**Figure 3.** Global monthly H$_2$ sources (top: a,b) and sinks (bottom: c,d) for 2015 (left: a,c) and 2016 (right: b,d), calculated using the GEOS-Chem baseline simulation. Both sources and sinks are aggregated by latitudinal band as follows: red - High Northern Hemisphere (HNH) north of 45°N, orange - Lower Northern Hemisphere (LNH) from 15°N to 45°N, yellow - Tropics (TP) from 15°S to 15°N, light blue - Lower Southern Hemisphere (LSH) from 15°S to 45°S, dark blue - High Southern Hemisphere. The emissions (Emis) include anthropogenic, biomass burning and biogenic N$_2$ fixation sources. The chemical production (ChemProd) includes photochemical formation from formaldehyde and glyoxal. The chemical loss (ChemLoss) considers the reaction with OH.

We use the CSIRO measurements (Krummel et al., 2021b, c, d, e, f, g, h, a; **?**) to assess the H$_2$ seasonal cycles. Model biases
and other statistical metrics calculated are shown in Table S2 in the supplementary material. The supplement also includes a comparison of modelled and observed H$_2$ vertical profiles using aircraft measurements (Krummel et al., 2021i). These are shown in Figure S7 and are not discussed further here.

Figure 4 shows a seasonal spatial average of H$_2$ mixing ratios in surface air averaged over 2015 and 2016, with simulated values overlaid by the observations of CSIRO. Modelled mixing ratios at 500 hPa can be seen in Figure S6. As expected, in
each hemisphere, the H$_2$ mixing ratios are lower in the corresponding summer and autumn than in spring and winter months. This seasonal trend is driven by the soil uptake (Figure 3) and its relationship with soil temperature and moisture. During summer and autumn, the temperature conditions are optimal for the soil uptake of H$_2$, yielding lower concentrations of H$_2$ in surface air.

In the Northern Hemisphere, the mixing ratios of H$_2$ were highest during the December-January-February (DJF) and the
March–April–May (MAM) periods. Modelled H$_2$ was lowest in SON over Russia (∼400 ppbv), followed by North America (∼450 ppbv). The high modelled mixing ratios over China and Korea (∼600 ppbv) remained almost constant throughout the year, consistent with high emissions from anthropogenic combustion sources (see Figure 2). The season with the highest


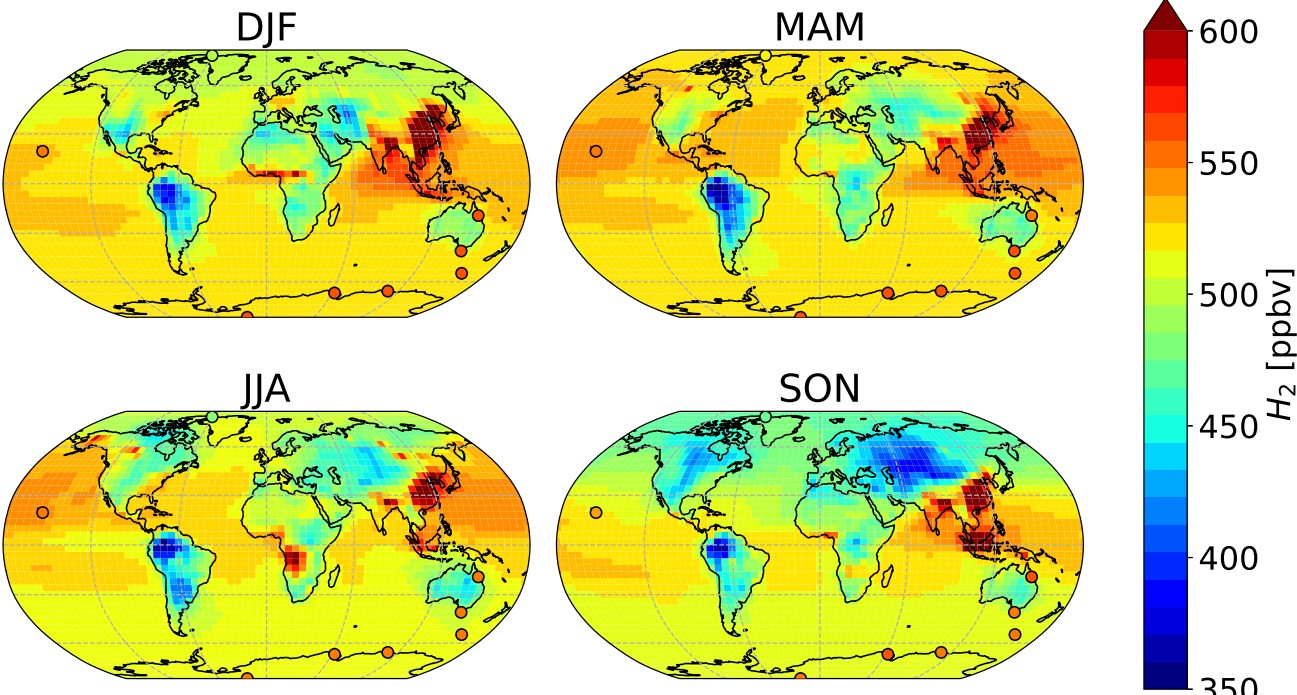

**Figure 4.** Average surface air $H_2$ mixing rations in each season as estimated by the GEOS-Chem baseline simulation (background), compared against average measured values from CSIRO (circles). Both modelled and observed values have been averaged over 2015-2016.

modelled estimates of $H_2$ over China and Korea is DJF. A similar trend in the estimated mixing ratios of CO implies that the anthropogenic emissions inventories used over this region are likely responsible for the high values of these modelled gases.

In the Southern Hemisphere, elevated $H_2$ mixing ratios on the order of 550-600 ppbv modelled over Africa and Indonesia in austral winter-spring (JJA and SON) coincide with the seasonal cycle of biomass burning emissions (Pak et al., 2003; Edwards et al., 2006). Throughout the year, the lowest $H_2$ mixing ratios globally are found in South America, in particular in the Amazon region. Over these regions, modelled mixing ratios are consistently lower than 450 ppbv. To our knowledge, there are no available measurements of $H_2$ for South America that could be used to evaluate the modelled mixing ratios there.

Observations are also lacking over most of the Middle East, parts of Asia, Africa and Australia. $H_2$ measurements over these regions would provide particularly valuable constraints in further modelling endeavours.

The visual comparisons in Figure 4 show that there are different biases in different locations, with notable underestimates at the the Southern Hemisphere observing sites.

While limited in spatial extent, the CSIRO data is well suited for evaluating modelled mixing ratios and seasonal patterns.

Figure 5 compares monthly mean modelled mixing ratios of $H_2$ against measurements at the sites from the CSIRO ensemble for 2015 and 2016 (Krummel et al., 2021b, c, d, e, f, g, h, a). At the two sites located in the Northern Hemisphere (Alert, ALT and Mauna Loa, MLO), GEOS-Chem was able to capture both the magnitude and the majority of the variability over the two





years. At the six sites in the Southern Hemisphere (Cape Ferguson, CFA; Cape Grim, CGO; Macquarie Island, MQA; Casey, CYA; Mawson, MAA and South Pole, SPO) the model captures the observed seasonality but is biased low by $> \sim 20$ ppbv. In
other words, the CSIRO measurements indicate a persistent low bias in modelled Southern Hemisphere $H_2$ mixing ratios. The measurements at sites like Cape Grim (CGO) represent baseline-selected (clean air masses) only. The model output analyzed was not filtered to match background conditions at some of the CSIRO sites, a condition that could be a minor contributing factor to some of the observed bias. Figure S8 shows that the baseline-selected in-situ data have differences of $\sim 3$ ppb and are in good agreement with baseline-sampled flask data. On the other hand, a misrepresentation of biomass burning emissions or
biased inter-hemispheric modelled mixing ratios are unlikely the cause of the difference between observations and predictions because of the good model performance shown in CO estimates. However, a revision of the mass fractions used to scale the emissions inventories of CO to $H_2$ is recommended. Figure S9 in the supplement shows that the Southern Hemisphere modelled bias is unique to $H_2$ and is not seen in CO. Given the large ocean area in this part of the world, underestimated ocean $H_2$ emissions are a possible driver of the bias. Improvements to ocean $H_2$ emission parameterisations with particular emphasis
on the Southern Ocean should be a priority for future model development.

Overall, our baseline model was able to capture the main features of observed spatial and seasonal variability of the $H_2$ reported by CSIRO (Krummel et al., 2021b, c, d, e, f, g, h, a). Combined with the fact that the simulated $H_2$ budget, burden, and lifetime are all consistent with previous estimates, the observational evaluation lends confidence to the suitability of our baseline model configuration. In what follows, we further adapt our baseline configuration to test the impact on tropospheric
$H_2$ of its generation from photolysis of aldehydes other than formaldehyde and glyoxal.

### 3.3   GEOS-Chem modelling results: global implications of $H_2$ production from aldehydes

The GEOS-Chem chemical mechanism includes nine aldehydes: formaldehyde, glyoxal, glycolaldehyde, acetaldehyde, methylglyoxal, methacrolein, hydroperoxyaldehydes (HPALD), dihydroperoxide dicarbonyl and a lumped species called RCHO representing other aldehydes with three or more carbon atoms. As mentioned previously, the standard mechanism already includes
direct $H_2$ production from photolysis of formaldehyde and glyoxal. Here we tested the impacts of the direct formation of $H_2$ from photolysis of the rest of the aldehydes in GEOS-Chem (with the exception of dihydroperoxide dicarbonyl as it was not present in the box modelling test).

Photolytic $H_2$ production from aldehydes was added to the existing standard chemistry mechanism using KPP embedded within GEOS-Chem. We assigned the 1% quantum yield found by Harrison et al. (2019) for acetaldehyde to the selected
aldehydes tested in GEOS-Chem, analogously to what was done for the box modelling (Section 2.1). The 1% from acetaldehyde was taken as the reference quantum yield to test given that measurements for the rest of the aldehydes are not available, but the energy barriers for the production of $H_2$ from aldehyde photolysis indicates that the dissociation channels are accessible (Rowell et al., 2021). For acetaldehyde, glycolaldehyde, HPALD and RCHO, a branching ratio on the existing photolysis channels was added to account for the primary production of $H_2$ in addition to the existing photolysis products. For methacrolein and
methylglyoxal, additional steps had to be taken as the current GEOS-Chem implementation of Fast-JX for methacrolein and methylglyoxal embeds the quantum yields in the provided "cross sections". For these species, new photolysis channels were



**Figure 5.** Seasonal cycle comparisons of H₂ mixing ratios at the eight sites from the CSIRO flask measurements for 2015 and 2016 (Krummel et al., 2021b, c, d, e, f, g, h, a). The dash line with circle markers shows the observed values and the continuous line with triangle markers shows the modelled values from the GEOS-Chem baseline simulation. The colors are the same as in Figure 3. For a comparison of the modelled and observed CO at the same sites see Figure S9.





created that separated the cross sections from the quantum yield. The cross sections for the two species were retrieved from Sander et al. (2020) and processed using the Fast-J v7.3c model, which covers 18 wavelength bins from 177 to 850 nm (Prather, 2015). The resulting binned cross sections and 1% quantum yield were then configured back into the customized version of

Fast-JX v7.0 used in GEOS-Chem Eastham et al. (2014). Beyond these changes to the photochemistry, all other sources and sinks were identical to those used in the baseline. We ran this modified version of the simulation with the new photochemistry from June 2014 to December 2016, again using the first six months as spin-up. This simulation will hereafter be referred to as the aldehyde photolysis scenario.

    Figure 6 shows the percentage difference in total tropospheric $H_2$ chemical production between the aldehyde photolysis

scenario and the baseline simulation. The increase in the chemical production of $H_2$ from the new photochemistry for aldehydes is widespread across the globe. Figure 6a shows that the increase in total column $H_2$ chemical production reached a maximum of ∼10%, with the biggest changes taking place over the Amazon. Forested regions in the African tropics, Indonesia, Papua New Guinea and northeast Australia show increases that ranged from 2% to 8%. At the surface (Figure S10a), the increase in $H_2$ chemical production was up to 14% with the same spatial distribution as seen in Figure 6a for the troposphere as a whole.

In the vertical profile (Figure 6b and 6c), the increases in $H_2$ chemical production extended to 700 hPa over the tropics. This increase well above the surface layer may be a result of the strong vertical transport in this region , with rapid transport of aldehydes from the surface to the mid-troposphere followed by their photolysis to yield $H_2$.

    The strongest response to the new aldehyde photochemistry is seen in regions with dense vegetation cover characterized by high isoprene emissions. The most relevant aldehydes for the formation of $H_2$ over densely vegetated areas are thus those

related to the oxidation of isoprene and of its primary products, methacrolein and methyl vinyl ketone. Of particular importance here are methylglyoxal and glycolaldehyde, products from the OH-initiated oxidation of both methacrolein and methyl vinyl ketone, which account for ∼79% and ∼49%, respectively, of the global sources of these aldehydes (Fu et al., 2008; Wennberg et al., 2018).

    We conducted additional model sensitivity simulations to compare the $H_2$ production from each of the new aldehyde sources

(e.g., excluding formaldehyde and glyoxal). From these sensitivity simulations, we find that methylglyoxal contributes approximately 91% to the enhanced tropospheric $H_2$ chemical production from these additional aldehydes (Figure S11a) while glycolaldehyde, methacrolein, and the other non-isoprene related aldehydes (acetaldehyde, HPALD and RCHO) collectively account for the remaining 9% (Figure S11b). These results imply that the most relevant aldehyde to include in global model simulations for the direct photochemical formation of $H_2$ is methylglyoxal. We note that the estimated methylglyoxal mixing

ratios in our simulations (Figure S12) are comparable to those modelled and reported by Fu et al. (2008). Fu et al. (2008) compared their modelled methylglyoxal estimates against available observations finding no systematic bias at land sites. Although Fu et al. (2008) used few northern midlatitudes locations to perform their comparison, the similarity between our methylglyoxal mixing ratios and the ones by Fu et al. (2008) gives us confidence in our modelled methylglyoxal and subsequent generation of $H_2$ from its photolysis.

Despite the substantial increase in $H_2$ chemical production associated with the new aldehyde photochemistry, the change in the tropospheric $H_2$ mixing ratios is very small, with a maximum change of 0.3% over South America as shown in Figure 7a.



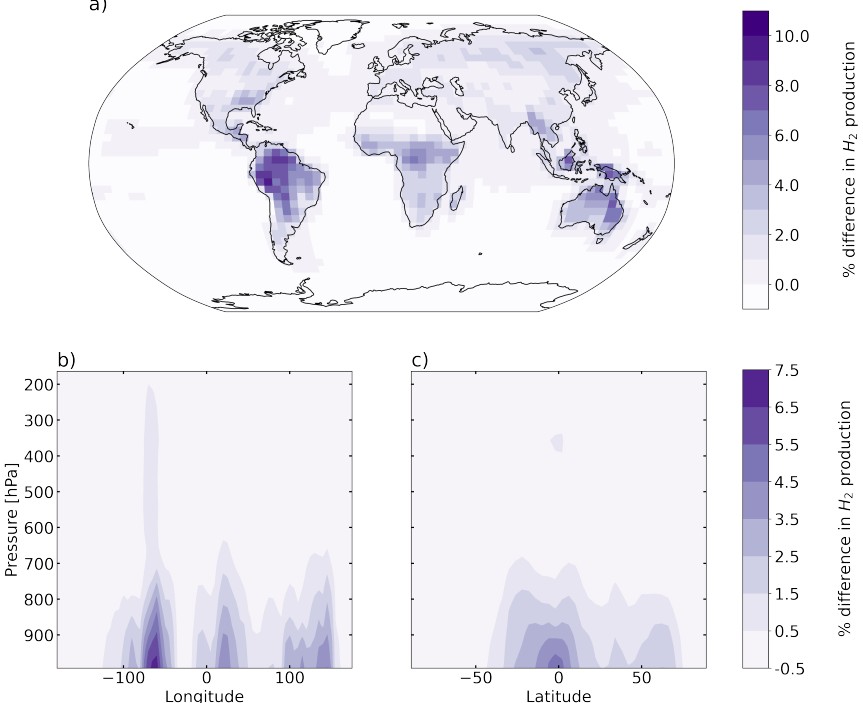

**Figure 6.** Percentage difference in tropospheric $H_2$ chemical production between the aldehyde photolysis scenario (with 1% $H_2$ quantum yield) and the baseline simulation, averaged over 2015 and 2016. Results are shown a) spatially (summed over the tropospheric column), b) as a function of altitude and longitude (summed over latitudes), and c) as a function of altitude and latitude (summed over longitudes).

This equates to a change at the surface over South America of ∼0.5% (see Figure S13a).As seen previously for the chemical production, the biggest changes occur over the tropics. The influence of long-range transport (facilitated by the ∼2-year $H_2$ lifetime) can be seen in the figure, with an increase of up to 0.2% in the $H_2$ mixing ratios over the oceans. Figure 7b and 7c

also show the injection of $H_2$ to higher levels in the troposphere, particularly in the tropics where the increase extends to 500 hPa, but as seen in the figure, the enhancement in the mixing ratios does not exceed ∼0.2%. At higher latitudes, the change is almost imperceptible, as expected by the lack of precursor aldehydes at those latitudes.

Figure 8 displays the absolute differences between the aldehyde photolysis scenario and the baseline simulation for chemical production (a), chemical loss (b), and soil uptake (c), all at the surface layer. The enhanced $H_2$ chemical production in the tropics

discussed previously (Figure 8a) is compensated by increased soil uptake of $H_2$ (Figure 8c), with both showing maximum values over the Amazon. Elsewhere the situation is much the same: the enhanced $H_2$ produced from aldehyde photolysis is largely deposited in the same locations, making the atmospheric enhancement of $H_2$ from aldehyde photolysis small. This implies that the increases in production had a tendency to occur in places and times where the loss rates are stronger than the global average. Although the effect is smaller than seen for the soil sink, the chemical loss of $H_2$ from reaction with OH (Figure

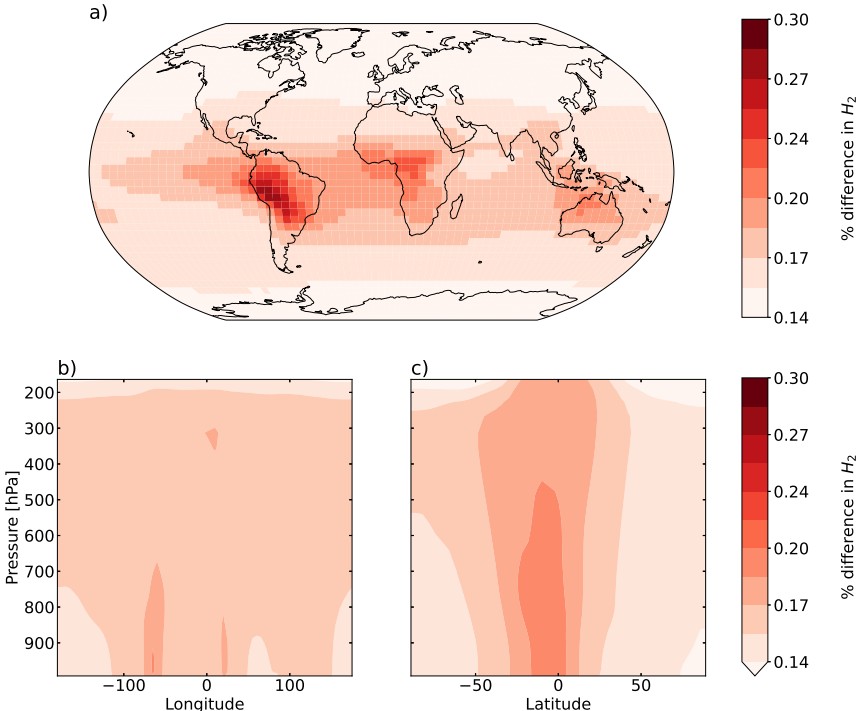

**Figure 7.** Same as Figure 6 but for average $H_2$ mixing ratios.

8b) also increases as expected in response to the enhanced production, further contributing to the balance between additional $H_2$ production and loss in the aldehyde photolysis scenario.

The tropospheric budget from the aldehyde photolysis scenario is shown alongside the budget from the baseline simulation in Table 2. Overall, the new photochemistry led to an increase in $H_2$ sources, sinks, and tropospheric burden compared to the baseline simulation, but all remained within the ranges reported by previous studies (Table 2). Summed over the global troposphere, the total increase in tropospheric $H_2$ chemical production from inclusion of direct $H_2$ production from newly-discovered aldehyde photolysis was only 0.98% for 2015 and 0.96% for 2016. Given the small changes in $H_2$ mixing ratios described above, there were no significant changes in model performance relative to observations at measurement sites. The low model bias observed in the Southern Hemisphere did not improve, which allows us to conclude that missing chemical sources are not likely to resolve the remaining uncertainties and biases in the modelled $H_2$ seen in the baseline simulation.

## 4 Summary and conclusions

Recent laboratory findings by Harrison et al. (2019) identified a previously unknown $H_2$ channel for acetaldehyde yielding $H_2$ at a 1% quantum yield. This finding by Harrison et al. (2019) was complemented by aldehyde ground state calculations that show that the direct $H_2$ channel is also possible for other aldehydes (Rowell et al., 2021). Here, we assessed the impact of the





**Figure 8.** Absolute differences at the modelled surface layer between the aldehyde photolysis scenario (with 1% quantum yield of $H_2$ from aldehydes) and the baseline simulation for a) $H_2$ chemical production, b) $H_2$ chemical loss, and c) $H_2$ uptake by soil. Units for all plots are Mg yr$^{-1}$. Note the differences in scale between the plots.





recently determined direct generation of $H_2$ from aldehyde photolysis using two photochemical models: the AtChem v1.2 box

model implementing the Master Chemical Mechanism MCM v3.3.1 and a modified version of the GEOS-Chem v12.5.0 3-D

chemical transport model.

We configured the box model at three sites (London, Cape Verde and Borneo) to explore the production of $H_2$ under distinctive atmospheric conditions and constrained each box model simulation with measurements. The standard MCMv3.3.1 considers 18 aldehydes and their corresponding reactions, with formaldehyde and glyoxal already including a $H_2$ channel. We

evaluated the generation of $H_2$ from the remaining 16 aldehydes in MCM by comparing a baseline simulation against 4 sensitivity scenarios each using different $H_2$ quantum yields (1%, 2%, 5% and 10%). The selected quantum yields for the sensitivity analysis were chosen based on experiments by Kharazmi (2018) that showed that methylpropanal has an 8% quantum yield for the $H_2$ channel. Our box model results allowed us to identify the aldehydes that are more likely to contribute to the $H_2$ production.

Excluding the contributions from formaldehyde and glyoxal, which remain the biggest photochemical $H_2$ sources, in an urban atmosphere, aliphatic aldehydes such as acetaldehyde and propanal contributed over 80% of the simulated photochemical generation of $H_2$ from aldehydes. Unsaturated olefinic aldehdydes and vegetation-related species like methylglyoxal, methacrolein and acrolein provided a collective contribution of less than 10%. The remaining minor contributions came from glycolaldehyde. In a marine atmosphere, results were similar, with acetaldehyde and propanal contributing to 90% of the $H_2$.

In an atmosphere over a tropical rainforest, the oxidation products of vegetation-emitted species (i.e., methylglyoxal, glycolaldehyde, methacrolein and acrolein) contributed to 81% of the $H_2$ produced. Based on the contribution at each modelled site, out of the 16 aldehydes tested with MCM, six were identified as the most relevant for $H_2$ production: acetaldehyde, propanal, glycolaldehyde, methylglyoxal, methacrolein, acrolein. Based on this finding from the box modelling, the global impacts of $H_2$ production from five of these aldehydes (excluding acrolein) were further investigated by using global atmospheric chemical

transport modelling.

We then developed a global GEOS-Chem simulation of $H_2$ by modifying the v12.5.0 code to simulate $H_2$ as an active species with tropospheric sources including direct emissions from anthropogenic combustion and biomass burning sources and photochemical production from formaldehyde and glyoxal, along with sinks from reaction with OH and soil uptake parameterised as a dry deposition flux. We simulated 2015-2016 (preceded by a six-month spin-up) and compared the results

against available measurements (Krummel et al., 2021b, c, d, e, f, g, h, a, i). The model performance analysis showed our new GEOS-Chem baseline $H_2$ simulation is able to reproduce the seasonal cycle of $H_2$ at the different measured sites. Model performance was better in the Northern Hemisphere than in the Southern Hemisphere, where a persistent low bias was present. An over-estimation of sinks and/or missing $H_2$ sources (particularly from the ocean) may explain the observed low model bias and should be investigated in future work. In the Northern Hemisphere, high estimates in East Asia seen for both $H_2$ and CO

are likely due to overestimates in anthropogenic emissions. Our simulated tropospheric budget of $H_2$ indicated a global burden of $\sim159$ Tg yr$^{-1}$ and a lifetime of $\sim2$ years, consistent with previous studies (see Table 2). Overall, the model performance was deemed satisfactory for use as a baseline simulation to compare to a modelled scenario with new $H_2$ production from aldehyde photolysis.



Six aldehydes were tested in GEOS-Chem, each with a 1% quantum yield channel for $H_2$: acetaldehyde, propanal (part of the lumped RCHO species), glycolaldehyde, methylglyoxal, methacrolein and hydroperoxyaldehyde (HPALD). We ran the model for the same two years as the baseline simulation (2015-2016), again with six-month spin-up, and compared the results against the baseline. We calculated a maximum increase in the tropospheric $H_2$ chemical production over tropical regions of ~10% as a result of the new aldehyde photochemistry. The spatial distribution of the newly produced $H_2$ correlated well with the distribution of aldehydes associated with isoprene oxidation: glycolaldehyde, methylglyoxal and methacrolein. Using additional sensitivity studies, we found that over 90% of the new chemical production could be attributed to methylglyoxal.

The ~10% increase in the chemical production of $H_2$ yielded an additional $\sim 3.6 \times 10^{-3}$ Mg yr$^{-1}$, an amount that was ultimately balanced by an increase in the chemical loss and soil uptake of $H_2$. The result of compensating sources and sinks in the aldehyde photolysis scenario was a maximum effective change of only 0.3% in the tropospheric mixing ratios, which was seen over South America. The minimum change in the tropospheric $H_2$ mixing ratios associated with the new photochemistry was 0.14%, found over the poles where aldehyde precursors are negligible. The additional $H_2$ source from aldehyde photolysis therefore did not improve the low model bias in the Southern Hemisphere seen in the baseline simulation. This means that other processes besides the photolytic loss of aldehydes are more likely responsible for the lingering discrepancies between model and measurements. These include both the emissions (natural and anthropogenic) and the soil uptake processes. Future work should focus in particular on improvements to anthropogenic emissions in areas with high bias, such as China and Korea, and ocean emissions in the Southern Hemisphere.

The implementation of the new aldehyde photochemistry in the two models yield consistent results, showing that the biggest changes in the chemical production of $H_2$ will occur for areas with a sizeable source of biogenic VOCs that can serve as precursors for the most relevant aldehydes identified in this work. Both models point to methylglyoxal as a potentially relevant photochemical source of $H_2$. The box model highlights an additional possible contribution from glycolaldehyde. The fact that methylglyoxal and glycolaldehyde make significant contributions to modelled $H_2$ production in our simulations is significant. While we did not distinguish here between different types of aldehydes, Rowell et al. (2021) explain that the aldehydes that most likely yield $H_2$ in the troposphere are those with a triple fragmentation (TF) channel with energies below 350 kJ mol$^{-1}$. Sufficiently low TF energy barriers have been calculated for both methylglyoxal (330 kJ mol$^{-1}$) and glycolaldehyde (229 kJ mol$^{-1}$) (Rowell et al., 2021). Glycolaldehyde has the lowest energy barrier for the TF channels of any of the aldehydes calculated by Rowell et al.. The glycolaldehyde TF energy barrier is even lower than that of propanal (295 kJ mol$^{-1}$), which has been shown to have a $H_2$ quantum yield of ~8% (Kharazmi, 2018). Thus, both methylglyoxal and glycolaldehyde are the aldehydes that are theoretically most likely to have a $H_2$ channel based on the calculations from Rowell et al. (2021) and (for glycolaldehyde at least) may have a $H_2$ quantum yield greater than the 1% tested in our simulations. The experimental determination of $H_2$ quantum yields from a TF channel for methylglyoxal and glycolaldehyde should therefore be prioritised. Further, our estimates for the contribution of methylglyoxal and glycolaldehyde to the photochemical generation of $H_2$ may well represent a lower limit of the true contribution.

Finally, our new GEOS-Chem $H_2$ simulation capability, including the new photochemistry for the direct generation of $H_2$ from aldehydes, provides a useful tool for other studies of $H_2$. This model can serve as a framework for interpreting historical



$H_2$ distributions and variability, improving process-level understanding of $H_2$ cycling, and testing future $H_2$ emission scenarios.
The latter will become increasingly important as plans to migrate to the $H_2$ economy begin to materialize.

*Code availability.* The modified version of GEOS-Chem v 12.5.0 used here for the baseline is available at https://github.com/mpperezp/atmH2_modelling

*Author contributions.* M.P.P.P performed all the simulations and data analysis. S.H.K. and J.A.F conceived and directed the project. D.B.M contributed with the oceanic $H_2$ emissions, set-up and analysis of the baseline simulation. H.Y provided the global dry deposition velocities of $H_2$. R.L.L and P.B.K contributed with the measurements used to compare the model and provided guidance to use the CSIRO measurements.
All authors contributed the drafting of the manuscript.

*Competing interests.* No competing interests are present

*Acknowledgements.* This work was funded by the Australian Research Council (DE200100549/DP190102013). This work was was undertaken with the assistance of resources provided at the NCI National Facility systems at the Australian National University through the National Computational Merit Allocation Scheme supported by the Australian Government (project m19).This research was undertaken
also with computer time on the computational cluster Katana supported by the Faculty of Science, UNSW Australia. The authors thank Dr. Lisa K. Whalley for her help relating to setting up and running the AtChem simulations and providing the information related with the constraints for the London box modelling. Past and present CSIRO GASLAB staff are thanked for their dedication to making high quality long-term measurements. CSIRO is thanked for the long-term institutional support of GASLAB and the CSIRO global flask network. The Australian Bureau of Meteorology, Australian Antarctic Division, Australian Institute of Marine Science, National Oceanic and Atmospheric
Administration, and Environment & Climate Change Canada are gratefully thanked for the long-term support for filling flasks and logistics at the field stations used in the CSIRO flask network. The determination of the dry deposition fields used in this work were supported by MEXT (JPMXP1020200305) as "Program for Promoting Researches on the Supercomputer Fugaku" (Large Ensemble Atmospheric and Environmental Prediction for Disaster Prevention and Mitigation).





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
