# Peer review of "Evaluating the contribution of the unexplored photochemistry of aldehydes on the tropospheric levels of molecular hydrogen $(H_2)$ ."

_Atmospheric Chemistry and Physics, 2021_

## Author Comment (AC1)

https://doi.org/10.5194/acp-2021-1052

**Evaluating the contribution of the unexplored photochemistry of aldehydes on the tropospheric levels of molecular hydrogen (H$_2$): Response to reviewers**

Maria Paula Perez-Peña, Jenny A. Fisher, Dylan B. Millet, Hisashi Yashiro,
Ray L. Langenfelds, Paul B. Krummel, Scott H. Kable

We thank both reviewers for the thorough reading of our manuscript and all the helpful comments. Their insight has contributed to the betterment of our work. In what follows we address each reviewer comments, the original comments are in black and our answers are indicated in blue. New text is indicated in **bold**.

**1 Reviewer 1**

The authors analyzed the contribution of aldehydes on the chemical production and tropospheric levels of $H_2$ using a box model and a 3-D atmospheric chemical transport model. The authors concluded that their results imply that the previously missing photochemical source is a less significant source of model uncertainty than other components of the $H_2$ budget. Overall, the paper is well written and well organized.

We thank the reviewer for the positive feedback about our paper.

**1.1 Major comment**

(a) My only concern is that the global model simulations were conducted with a resolution of 4x5. As the authors found in Section 2, the conditions over urban environments and regions with substantial vegetation are very different. Urban size is usually smaller than this scale. Can the model properly represent that with this resolution?

- We have added a clarification into the main text in Section 3.3 **"Even tough the box modelling showed a marked difference on the aldehydes that produce $H_2$ between urban and densely vegetated environments, the global model simulations are not intended to capture the fine-scale detail of such regions. Aiming at estimating how the aldehyde photochemistry compares to other $H_2$ global sources, the coarse resolution used in the baseline is maintained to test the reactions."** The 4x5 resolution used here is sufficient for the purpose highlighted now from lines 363 to 366. **"Even tough the box modelling showed a marked difference on the aldehydes that produce $H_2$ between urban and densely vegetated environments, the global model simulations are not intended to capture the fine-scale detail of such regions. Aiming at estimating how the aldehyde photochemistry compares to other $H_2$ global sources, the coarse resolution used in the baseline is maintained to test the reactions."** Also, the applied resolution is similar to other global $H_2$ budget simulations, such as the one by Derwent et al., (2021), that used a resolution of 5x5. Detailed accounting of the budget in urban environments would require a higher resolution model as noted by the reviewer, but that is beyond the scope of the work presented here.

**1.2 Minor comments**

(1) Line 95: "The the" to "Then the"?

- The correction in line 95 has been made.

(2) Line 216: Here it is "GFED4" but later it was denoted "GFEDv4s". Please make it consistent.

- The abbreviation for the Global Fire Emissions Database v4 was standardized as GFEDv4s throughout the text.

(3) Line 345: Change "difference between observations and predictions" to "difference between observations and model simulations"?

- As suggested by the reviewer, the sentence has been changed.

(4) Figure 1. The colors are hard to differentiate especially the orange colors.

- Figure 1 has been updated to help the reader differentiate the colors better.

**2   Reviewer 2**

In this study the authors estimate the contribution of aldehyde photolysis on the production of $H_2$.

Using a chemical transport model, the authors conclude that this recently identified source makes a very small contribution to the overall $H_2$ budget.

This analysis is performed using a newly-developed simulation of $H_2$ in the GEOS-Chem model.

The findings are interesting but require additional analysis for publication in ACP.

**2.1   Major comments**

(a) Section 2

(1) Given the limitations of the box model, the interpretation of the simulated $H_2$ concentration is quite challenging as pointed out by the authors. I suggest to focus on the production rates [i.e., remove lines 125 to line 146]

- Following the suggestion from the reviewer we have placed the interpretation of the mixing ratios into the Supplementary material.

(2) Fig. 1. Please indicate the $H_2$ production associated with aldehyde photolysis at each site and the time period considered. What is the production of $H_2$ from the photolysis of formaldehyde and glyoxal at these sites? If the production from aldehydes is very small relative to that from formaldehydes and glyoxal, the authors need to clarify why they expect a different result from the global model.

- The $H_2$ rates of production at each site are now included in the caption of Figure 1. We also now explicitly state that the production of aldehydes was small relative to that of formaldehyde and glyoxal from lines 166 to 168 **"The aggregated rates of production from the tested aldehydes here was less than 1% of the total rate of production at each modelled site, with formaldehyde and glyoxal remaining the main photochemical sources of $H_2$ in the box model simulations."** We did not anticipate a different result with the global model. However, as explained in lines 171-175, we turned to the global modelling to be able to appropriately quantify the impact by including other processes that are relevant to the $H_2$ budget, accounting for spatial variability in environments, and extending the simulation time to allow the model to reach steady state.

(3) It's unclear why the authors restrict themselves to these 3 sites. Would it be possible to perform the same type of analysis using data from NOAA/NASA airborne intensive campaigns (acetaldehydes and glycoladelhydes are often measured)? This would help better constrain the importance of this process.

- The goal of the work was to determine the relevance of the so-far unaccounted for photochemistry of some aldehydes and their role in the chemical production of $H_2$. This was clarified from lines 83 to 87 **"The box modelling aimed to determine which aldehydes contribute meaningfully to the primary chemical production of $H_2$ in different types of environments. We choose 3 indicative sites to explore distinctive environments, each with different expected mixing ratios of aldehydes (urban, pristine oceanic and pristine forested). The direct production of $H_2$ from the photolysis of aldehydes evaluated with AtChem further helped to configure the global model (see section 3.3)."** While simulating more observation campaigns would give a more nuanced view of the distribution of $H_2$-forming compounds, it would not fundamentally change our understanding of the chemistry or of the most important aldehydes to include in our simulations.

(b) Section 3

(1) Emission inventory.

The authors use a constant emission factor to convert anthropogenic CO emissions to $H_2$ emissions. Instead Ehhalt et al (2010) recommended using different emission factors for transportation.

Similarly, Akagi (2011) et Andreae (2019) reviews provide biome-specific emission factors for $H_2$. It's unclear why the authors didn't use these.

Indeed, the Akagi estimates are already used by GFED4s.

- As suggested by the reviewer, we modified the scaling factors used for anthropogenic fossil fuel emissions making a distinction for automobile emissions. The new scaling factor for automobile emissions was 0.036 g $H_2$ / g CO as summarized by Ehhalt. Added to this, we changed the use of a single scaling factor for biomass burning emissions from CO emissions, and instead applied the biome-specific emission factors from Akagi et al., (2011) and Andreae (2019), as also done by Paulot et al., (2021), to the GFEDv4s implementation in GEOS-Chem. With this modifications we re-ran both the baseline and the aldehyde photolysis scenarios. The changes are reflected in the budget estimates and the values in the text have been updated (see section 3.2).
  The resulting changes were small, and both our original and updated budgets are within the range found in previous estimates. For example, Paulot et al. (2021) determined a range for biomass burning emissions of 7.3-12.6 Tg $H_2$/yr over a 1995-2014 simulation. Our original estimates for 2015-2016 with a single CO scaling factor were 6.5-9.2 Tg $H_2$/yr. These values have been updated to 7.6-9.2 Tg $H_2$/yr (see Table 2 and Figure 3). All figures and the text have been updated to reflect the changes, but we note that there is no change to any of our conclusions following this modification.

(2) Deposition velocity

  (1) The authors need to discuss how the Yashiro model differs from the one presented by Ehhalt (2013). The Ehhalt model parameterization was used in two recent studies (Bertagni (2021) and Paulot (2021)). It would be interesting to test the model against observations collected at Harvard Forest by Meredith (2017)

  - We have now added a clarification on how the model used by Yashiro et al., (2011) differs from the one used by Ehhalt et al., (2013) from lines 229 to 233 **"The parameterisation to derive the dry deposition velocity for $H_2$ used by Yashiro et al., (2011) implements the same variables as the ones used by the Ehhalt et al., (2013). The Ehhalt et al., (2013) parameterisation, applied in other recent modelling studies Paulot et al., (2021), differs from the one by Yashiro et al., (2011) in that the latter considers the diffusivity in the soil to be uniform from the soil surface to a sufficient depth, while Ehhalt et al., (2013) uses two different soil diffusivities."** We agree that it would be interesting to compare simulated dry deposition models against the $H_2$ fluxes measured by Meredith et al., (2017), but this is beyond the scope of our work which focuses on the untested aldehyde photochemistry at global scales (and does not attempt to update or evaluate dry deposition schemes).

  (2) It's unclear why the authors do not calculate vd($H_2$) dynamically in the model using the parameterization described by Yashiro and soil moisture/temperature available from reanalysis (see Bertagni (2021), for instance). This would provide a significant improvement over the approach used by Price (2007) that solely (but interactively) accounts for the impact of snow cover and temperature.

  - While we would have preferred to calculate vd($H_2$) dynamically in the model, this is not possible in the GEOS-Chem CTM. Unlike in the examples cited by the reviewer (climate models that integrate with a land model Paulot et al., (2021)), GEOS-Chem does not have access to several of the soil variables required by the Yashiro et al., (2011), Ehhalt et al., (2013) and Bertagni et al., (2021) parameterisations, such as soil porosity and depth of soil active layers. We added a clarification to this in the main text from line 225 to 228 **"The integration of an online $H_2$ dry deposition calculation from other studies (Yashiro et al., 2011; Ehhalt et al., 2013; Bertagni et al., 2021) was not performed given that the algorithms require soil variables (e.g. soil porosity and depth of soil active layers) that are not available in our model."**. The implementation of the dry deposition algorithms dynamically in GEOS-Chem would require extensive modification of the model. For this reason we do not calculate the vd($H_2$) dynamically. We note that the offline vd($H_2$) fields used here still represent a significant improvement from the approach used by Price et al., (2007).

  (3) I am not sure that I understand the benefit of using daily vd($H_2$) derived for a different period from the one considered here. This will not account for the significant swings in $H_2$ removal associated with soil moisture (see Bertagni (2021) for instance).

  - We use the daily dry deposition velocities from Yashiro et al., (2011) from 1997 to 2005 to construct a multi-year mean climatology that provides a reasonable estimate of the seasonal changes associated with different soil moisture and temperatures. We have added a comment to this effect to the main text from line 235 to 237.
    **"We used the dry deposition velocities from Yashiro et al., (2011) to create a**

**climatology for an "average" year with daily temporal resolution to represent typical seasonal variability, which was then used in our 2014-2016 simulations"** While we could have used monthly means, that would have introduced an assumption that step changes occur at the same time each year (e.g. end of each month). Averaging the daily values across years provides for smoother transitions that are more representative of the real interannual variability. We do not expect to capture specific fluctuations associated with changes in temperature or soil moisture, and indeed do not show results at sub-monthly resolution

(3) Impact of hydrogen.

The authors need to describe how the improved representation of $H_2$ in GEOS-Chem impacts the lifetime of $CH_4$, the tropospheric and stratospheric $O_3$ budget, and the stratospheric $H_2O$ budget. As detailed in many studies (Derwent et al. (2000,2020), Field (2021), Paulot, (2021), Vogel (2012))), these are critical to understanding the indirect impact of $H_2$ on radiative forcing.

- The goal of this paper is to evaluate the impact of the changes in chemistry on the $H_2$ budget, not to understand the role of $H_2$ on the climate system. Evaluation of changes in follow-on parameters such as $CH_4$ and $O_3$ is beyond the scope of this work. In any case, the change in $CH_4$ cannot be determined from our simulations as the GEOS-Chem Standard simulation used is nudged to $CH_4$ observations. The change in the $O_3$ fields is very small with differences <1%.

(4) Evaluation Why aren't airborne observations discussed (Fig. S7)? This is a unique dataset and the model does show significant biases that should be discussed.

- The airborne dataset is discussed very briefly in the supplementary material, this discussion has been moved into the main text. To note is the fact that the comparison showed is a climatological one. The measurements used were performed from 1991 to 2000.

**2.2 Minor comments**

(1) Fig. 1 Please use IUPAC names for chemicals.

- As suggested by the reviewer, the IUPAC names were included in the Table S1 to allow for some of the more common names (as glycolaldehyde) to be consistent throughout the text.

(2) Fig. 3 is very difficult to read. Please use different colors for model and observations.

- Figure 3 shows sources and sinks for the two modelled years and makes no comparison against observations. The caption of the Figure has been updated to avoid further confusion.

(3) Are the authors using the results of the box model to select the aldehydes used in GC?

- We used the box model result to determine which aldehydes could be tested in GEOS-Chem, this is now clarified from line 360 to 362 **"Here we tested the impacts of the direct formation of $H_2$ from photolysis of the rest of the aldehydes in GEOS-Chem (with the exception of dihydroperoxide dicarbonyl as it was not present in the box modelling test). This was supported by the findings made in section 2.2, that showed the more relevant aldehydes to the photochemical formation of $H_2$."**

(4) A fairly recent $H_2$ budget was provided by Paulot (2021), which could be added to Table 2.

- The $H_2$ budget estimated by Paulot et al., (2021) has been added to Table 2.

(5) The AGAGE network provides $H_2$ observations at Mace Head that should be considered.

- The observations at Mace Head in Ireland have been included in the evaluation. The addition can be seen in Figures 4, 5, S9 and in Table S2.

(6) The lifetime of $H_2$ is 2.5 years. Isn't a 6-month spin up much too short?

- We set the initial $H_2$ mixing ratios to 530 ppb based on observations to enable the model to stabilise more quickly, and found that the model reached steady state after a 6-month spin-up. We have now clarified this in the main text.

(7) Fig. 5. Are these dry mixing ratios?

- The mixing ratios in Figure 5 are dry mixing ratios. We have added this to the figure caption.

(8) Line 348. The model seems to be biased low everywhere. Isn't $CH_4$ oxidation another possible culprit.

- $CH_4$ is constrained to the measurements in our GEOS-Chem model simulation. The model biases in the Southern Hemisphere are much larger than in the Northern Hemisphere, which suggests they are more likely attributable to low oceanic emissions, as we would expect a $CH_4$ oxidation bias to be more uniform. An addition in the main text in in lines 349 and 350 has been done **"Another possible source of model bias is the oxidation of $CH_4$. However, we do not expect it to represent the main source of the bias because $CH_4$ is constrained to observations at the surface in our simulations."**

(9) Fig. 2a. I cannot distinguish between brown and red

- Figure 2a has been updated by changing the color for the anthropogenic emissions to purple and biomass burning emissions to red to aid the reader.

(10) Fig. 5. Please show the impact of aldehydes on the simulated $H_2$ profile.

- Figure 5 has been modified to show the modelled scenario with $H_2$ photochemical production from aldehydes.

---

## Author Response (AR2)

https://doi.org/10.5194/acp-2021-1052

**Evaluating the contribution of the unexplored photochemistry of aldehydes on the tropospheric levels of molecular hydrogen ($H_2$): Response to reviewers**

Maria Paula Perez-Peña, Jenny A. Fisher, Dylan B. Millet, Hisashi Yashiro, Ray L. Langenfelds, Paul B. Krummel, Scott H. Kable

We thank the reviewer for the second revision to our manuscript, which has allowed for further improvement of our work. Below we include the original comments by the reviewer in black and our answers indicated in blue. New and modified text is indicated in **bold**.

**1    Reviewer 2**

**1.1    General comments**

(a) Line 253: Most of the study is devoted to improving the representation of $H_2$ in the GEOS-Chem model. As a result, I am still puzzled that vd($H_2$) is not calculated dynamically. As noted by the authors, it is the single largest sink of $H_2$ (>70%) and many (all?) recent models of atmospheric $H_2$ include such representation (see also Sanderson (2003), Pieterse (2011), Bousquet (2011)).

My understanding is that GEOS-Chem relies on GEOS-FP or MERRA-2 for inputs, both of which do include soil moisture and soil porosity. If I understand properly this recent study (https://www.nature.com/articles/s41597-020-0488-5), these fields are already used to estimate soil NO emissions, which would make it seem as if it should not be too cumbersome to implement the Yashiro model in GEOS-Chem. However, the authors mention in their reply that this is not possible. Could they elaborate?

- To clarify, the main focus of the work is not on improving the $H_2$ simulation in GEOS-Chem but on testing the impacts of the aldehyde photochemistry (as can be seen in the title, abstract, and introduction). To do so required us to have a reasonable $H_2$ simulation baseline that we could compare against our sensitivity simulations with the new baseline. As $H_2$ has not been included in any modern version of GEOS-Chem, we devoted a section of the paper to describing and evaluating our implementation.

- Turning specifically to the dry deposition term, we use the vd($H_2$) calculated externally by Yashiro, as the dynamic calculation requires soil variables that are not readily available in the meteorological fields used in GEOS-Chem. We note that GEOS-Chem does not use the GEOS-FP/MERRA-2 products directly, but instead requires post-processed versions of these products to make them compatible with GEOS-Chem. In the post-processing, which is performed by the GEOS-Chem Support Team before providing inputs to the community, only a subset of GEOS-FP/MERRA-2 variables are archived (http://wiki.seas.harvard.edu/geos-chem/index.php/List_of_GEOS-FP_met_fields). Unfortunately, several of the soil variables are not included in this subset, including those we would need to implement the Yashiro parameterisation dynamically. The soil parameters required to implement the dynamic calculation include: soil moisture, soil temperature and soil porosity. We describe below the availability of each of these variables for use in GEOS-Chem, along with comments on what is used in the soil NOx parameterisation in the paper referenced by the reviewer:

  – Soil moisture: The soil NOx emissions calculation in GEOS-Chem uses top soil wetness. The variable used (GWETTOP) corresponds to the ratio of the volumetric soil moisture to soil porosity. Neither soil moisture nor soil porosity are provided directly. This means that in order to use the top soil wetness from GEOS-FP as soil moisture, we also require the soil porosity.

  – Soil porosity: The original meteorological products include a porosity variable. However, this variable is not included in the post-processed version of the datasets that are used as input to GEOS-Chem. It is not used in the soil NOx emissions calculation.

  – Soil temperature: Soil temperature is available in the raw products but not in the products processed for input to GEOS-Chem. The soil NOx code uses the 2-m air temperature as a proxy

for this variable. While it would be possible to use the same approximation for the $H_2$ deposition calculation, it would add significant uncertainty to the dynamic calculation of the vd($H_2$) that should be evaluated appropriately.

- Considering that the calculation of vd($H_2$) builds on these three core parameters, we cannot implement the existing scheme with what is currently available in GEOS-Chem. We have clarified this in the manuscript, and also highlighted the importance of future work exploring the possibility of a dynamic sink term, from line 224 to line 229: "The integration of an online $H_2$ dry deposition calculation from other studies (Yashiro et al., 2011; Ehhalt and Rohrer, 2013; Bertagni et al., 2021) was not performed given that the algorithms require soil variables (e.g., soil porosity, **soil moisture, soil temperature** and depth of soil active layers) that are not available in **the post-processed GEOS-FP meteorological fields used as input to GEOS-Chem. However, the variables are available in the raw GEOS-FP (and MERRA-2) dataset. Future work should explore processing these variables for use in GEOS-Chem and implementing online soil uptake into the model.**"

(b) Line 257

(A) Please include the equation for vd. This would make this section easier to follow

- We have included the equation in the supplementary material.

(B) I believe that there are more differences between the two models (e.g., different sensitivity to temperature, moisture, ...). Please clarify.

- We have expanded on the differences between the two and moved the discussion to the Supplementary Material Section 2, *Dry deposition velocity calculation in Yashiro et al., 2011* In addition to including the relevant equations (as requested above), the section now reads as follows: **"The parameterisation to derive the dry deposition velocity for $H_2$ used by Yashiro et al., 2011 implements the same variables as that used by Ehhalt and Rohrer, 2013. The Ehhalt and Rohrer, 2013 parameterisation, applied recently by Paulot et al., 2021, differs from the one by Yashiro et al., 2011 in that the latter considers the diffusivity in the soil to be uniform from the soil surface to a sufficient depth (because the diffusivity is within the first layer of the parent land model.), while Ehhalt and Rohrer, 2013 use two different soil diffusivities. Further, the biological activity (uptake rate $k$), soil moisture and soil temperature dependencies are also different between Yashiro et al., 2011 and Ehhalt and Rohrer, 2013. Yashiro et al., 2011 follow the variation of the biological activity from Smith-Downey et al., 2006, while Ehhalt and Rohrer, 2013 rely on the dependencies from the reanalysis performed by Ehhalt and Rohrer, 2011. Also, the thickness of the inactive layer used by Yashiro et al., 2011 is considered to be uniform (with a value of 0.7 cm) while Ehhalt and Rohrer, 2013 provide values that are a function of the average volumetric soil water content."**

- We also refer to this section of the supplement in the main text when we describe our implementation of the dry deposition sink.

(C) Most importantly, although Yashiro and Ehhalt start from the same equation, if I am not mistaken they seem to arrive at a different result. Take delta=0, equation (13) of Ehhalt implies vd=sqrt(k Theta D) while equation (11) of Yashiro implies vd=Theta sqrt(k*D). I believe the former is correct. At any rate this should be clarified.

- We have added a brief discussion of this difference to the Supplementary Material (new Section 2)" **"Even though both models start from equation 1 (where $Fs$ is the flux and $C$ is the mass concentration of $H_2$) to derive the dry deposition velocity as a function of the $H_2$ flux and diffusivities, the two models differ when the thickness of the inactive layer $\delta = 0$ as a consequence of the difference in the definition of the flux.**

$$Vd = \frac{Fs}{\rho C} \tag{1}$$

**Ehhalt and Rohrer, 2013 used equation 2, where Ms corresponds to the $H_2$ mixing ratio in the soil air.**

$$Fs = -Ds\rho \frac{\delta Ms}{\delta z} \tag{2}$$

**On the other hand, Yashiro et al., 2011 used the equation 3**

$$Fs = -Ds\rho\theta_a \frac{\delta Ms}{\delta z} \tag{3}$$

> **This equation originated from equations (2), (3) and (4) in Yonemura et al., 2000. The version used by Yashiro et al., 2011 means that the gradient of the gas concentration between two layers is determined not only by the mixing ratio, but also by the air-filled porosity."**. Note that Eq. 6 in Yonemura et al. (2000b) misrepresented the porosity (as is not the combination of Eq. 3 Eq. 4 ), and so Yashiro et al. considered that $V_d = \theta\sqrt{kD}$ more relevant.

(c) Line 382: I don't follow the argument. If the simulated OH is biased, wouldn't you underestimate the flux of $H_2$ from CH4?

- We have clarified this in the manuscript in lines 359 to 361: **"We do not expect the oxidation of $CH_4$ to be a possible source of our $H_2$ model bias because $CH_4$ is constrained to observations at the surface in our simulations."**

**1.2 Minor comments**

(a) Line 380. Did you try to estimate the oceanic flux of $H_2$ that would be required to reduce the model bias?

- We did not attempt to determine the required oceanic flux to reduce the model bias observed for $H_2$, as this is beyond the scope of our work (focused on the photochemistry). We have clarified this in lines 356 to 359 "Given the large ocean area in this part of the world, underestimated ocean $H_2$ emissions are a possible driver of the bias, **although we did not estimate the magnitude of the emissions that is required to overcome such bias.** Improvements to ocean $H_2$ emission parameterisations with particular emphasis on the Southern Ocean should be a priority for future model development."

(b) Line 214. How do you assess that the model is indeed spun-up? Did you run longer spin-ups? If I am not mistaken, your IC is the same throughout the atmosphere. Given $H_2$ long lifetime, I would expect that it would take significantly longer to achieve a proper spin-up.

- We tested the impact of different spin-up times for January 2015 using longer spin-up times (18 months and 2.5 years, compared to our original 6 months). Over background regions, the difference between using our 6-month spin-up and longer spin-up times was less than 0.5%, and so we consider that our spin-up time was sufficient. We have clarified this in lines 187-189 **"Tests with 18-month and 2.5-year spin-ups showed differences in $H_2$ mixing ratios were smaller than 0.5%, showing that the six-month spin-up was sufficient."**